REGISTERED REPORT PROTOCOL

# Knowledge, behaviours and attitudes towards Evidence-Based Practice amongst physiotherapists in Poland. A nationwide cross-sectional survey and focus group study protocol

**Maciej Płaszewski** [1]⊙*, **Weronika Krzepkowska** [2]⊙, **Weronika Grantham** [3]⊙,
**Zbigniew Wroński** [4], **Hubert Makaruk** [3], **Joanna Trębska** [5]

1 Department of Rehabilitation, Faculty of Physical Education and Health in Biała Podlaska, The Józef Piłsudski University of Physical Education, Warsaw, Poland, 2 Department of International Cooperation, The Polish Chamber of Physiotherapists, Warsaw, Poland, 3 Faculty of Physical Education and Health in Biała Podlaska, The Józef Piłsudski University of Physical Education, Warsaw, Poland, 4 Department of Rehabilitation, Warsaw Medical University, Warsaw, Poland, 5 Institute of Econometrics, University of Łódź, Łódź, Poland

⊙ These authors contributed equally to this work.
\* maciej.plaszewski@awf.edu.pl

## Abstract

### Objectives

Evidence-Based Practice, EBP, is recognised, along with team work and patient-centred care, as a core competency in contemporary healthcare. However, dissemination and implementation of EBP remains problematic and is dependent on various external and internal factors, from personal through institutional to systemic, factors, with specific characteristics for different professions, contexts and settings. Knowledge, behaviours, attitudes, as well as facilitators and barriers towards EBP amongst physiotherapists, have been widely explored worldwide, but never in Poland. This study is part of a nationwide project, including dissemination actions addressing EBP in physiotherapists registered in Poland. Our purpose is to explore the issues of knowledge, behaviours, experiences, and attitudes of physiotherapists in Poland towards EBP. Descriptive research studies are warranted before analytical investigations and dissemination activities are conducted.

### Methods

We plan to conduct a quantitative, cross-sectional study–an online survey amongst the total population of physiotherapists registered in Poland to assess knowledge, behaviours and use of EBP (Study 1), and a qualitative study to allow physiotherapists to voice their opinions and to explore their experiences and attitudes towards EBP (Study 2). The EBP[2] questionnaire, Polish validated translation, will be used for Study 1, in a web-based survey. A focus group approach will be applied for Study 2, with purposive sampling to achieve a

**Data Availability Statement:** All relevant data from this study will be made available upon study completion.

**Funding:** The study is supported by the Ministry of Education and Science, Poland, within the project Societal Duty of Science, grant no SONP/SP/461408/2020, the Józef Piłsudski University of Physical Education in Warsaw, Poland, and the Polish Chamber of Physiotherapists. The funders have no role in study design, data collection and analysis, decision to publish, or preparation of the manuscript.

**Competing interests:** The authors have declared that no competing interests exist.

representative picture of physiotherapists with respect to setting, specialty, seniority, educational degrees, and age. We will follow an inductive approach, using topics rather than questions.

## Results

We will present the results of the studies separately, as typically presented in relevant study types: Study 1 will be reported addressing the domains and items of the EBP[2], in relation to the independent variables, and Study 2 will be discussed using the themes and illustrative quotes analyses.

## Discussion

We are aware that significant non-response, spin and Hawthorne effect may potentially bias our findings.

## Introduction

It is now thirty years since the Evidence-Based Medicine Working Group coined the term and set the principles of the paradigm shift of the practice and teaching of medicine [1]. Since that time, evidence-based medicine, EBM, with terminology then evolved to evidence-based practice, EBP, and other corresponding terms, has become a principle of healthcare [1–4]. Among many definitions, the World Confederation for Physical Therapy defines EBP as "an approach to practice wherein health professionals use the best available evidence from systematic research, integrating it with clinical expertise to make clinical decisions for service users, who may be individual patients/clients, carers and communities/populations (. . .) [which] involves complex and conscientious decision-making based not only on the best available evidence but also on patient/client characteristics, situations, and preferences (. . .)" [5].

Healthcare practitioners and other professionals are expected to implement research evidence in daily practice to improve the quality of care and patient outcomes [2–8]. Implementation of EBP is recognised not only as an improvement in understanding and optimising the process of care in terms of knowledge and abilities, but as a moral obligation. It is unethical to deliver suboptimal, ineffective, dangerous, or cost-ineffective interventions to patients and clients, as well as the unethical practice of conducting unneeded and flawed research [2, 7, 8].

EBP has become a necessary competency for rehabilitation [9] and physiotherapy practice [3, 4, 6]. Important initiatives, such as the World Confederation for Physical Therapy Policy Statement on Evidence based Practice [3], the Sicily statement on evidence-based practice [4], the Physiotherapy Evidence Database, PEDro [10], and focus on EBP in physiotherapy curricula [6], have grounded EBP as a core of physiotherapy education and practice.

## Why it is important to do this study

Further developments and evolution of the EBP movement, such as shifting from the traditional hierarchy of evidence to the GRADE approach, stressing patient values and shared decision-making in the process of EBP, and introduction of systematically developed, evidence-based clinical practice guidelines [2, 11], have been recognised in the standards of physiotherapy education and practice [5, 6]. Nonetheless, the barriers and difficulties to the implementation and dissemination of the process of EBP [11–13], also regard physiotherapy [14, 15], with

significant discrepancies across countries and contexts [16–18], and specific features of physio-therapists amongst other healthcare professionals [13, 19, 20].

Knowledge, skills, beliefs, and attitudes towards EBP (the EBP profiles [19, 21]) have been studied amongst physiotherapists worldwide [14, 15, 19, 22–27] since the first study addressing this problem was published in 2003 [28].

In contrast, to the best of our knowledge, no such research has been completed in Poland. There are unique contextual factors on EBP profiles and, more widely, on EBP culture amongst physiotherapists in Poland, so that descriptive research studies are needed first to allow further analytical investigations and dissemination activities regarding this crucial issue.

### Objectives

The overall aim of the project is to improve and facilitate the process of the dissemination of EBP in Polish physiotherapy. We need data on EBP profiles of physiotherapists in Poland to follow with further dissemination and implementation steps [5, 7–9]. Therefore, the specific objectives of the presented studies are:

- to assess EBP profiles of physiotherapists registered in Poland,

- to allow physiotherapists to voice their opinions and explore their experiences and attitudes towards EBP,

- to explore the association of a range of contextual factors and characteristics with their EBP profiles, and

- to investigate their experiences as regards the facilitators and barriers towards EBP in their everyday practice.

## Methods

The procedure comprises of two complementary and corresponding investigations–a quanti-tative, nationwide cross-sectional study (**Study 1**), and a qualitative, focus group study (**Study 2**). We describe the studies in subsequent paragraphs. The flow chart of the procedure is pro-vided as S1 File.

## Quantitative study (Study 1)

### Objectives

This is a nationwide study of EBP profiles–knowledge, attitudes, and use of EBP–amongst physiotherapists registered in Poland.

### Methods

**Design.** This will be a cross-sectional, online survey, addressed to the whole population of physiotherapists registered in the Polish National Registry of Physiotherapists. We will apply the Evidence-Based Practice Profile Questionnaire (EBP$^2$Q) [21], validated Polish version [29]. Personal, professional and demographic characteristics will be collected using additional ques-tions in the survey, as well as the data available through the National Registry.

**Setting and participants.** All registered physiotherapists will be invited to participate in the study. The criterion is the record from the National Registry, i.e. confirmation that a per-son has the legal status of a registered physiotherapist in Poland. Therefore, registered physio-therapists located in Poland, regardless of their nationality or country of origin, as well as Polish physiotherapists listed in the Registry, but living outside Poland, meet this formal

eligibility criterion. We will also invite people who will have obtained the status of a registered physiotherapist within two weeks of the first dispatch of the study. At the submission of this protocol report, the Registry comprised 70,052 records. The Registry is administered by the Polish Chamber of Physiotherapists, KIF. The project, including the study, is conducted in partnership with KIF so that current e-mail addresses and other contact details of all eligible participants are located and accessible.

In Poland, most of the registered physiotherapists are graduates of physical therapy education at master's or bachelor's degree studies. Physiotherapists, who obtained their vocational training prior to 2015, are also eligible for registration. Currently, the Polish entry-level physiotherapy education programme comprises of five years of master's studies. There is currently neither vocational training nor first-stage (bachelor) academic physiotherapy undergraduate education provided in Poland.

*Sociodemographic characteristics*. As the original demographic variables of the EBP$^2$Q were not validated in the Polish translation [29], and as we have specific needs for our study, we will supplement the original EBP$^2$Q sociodemographic content form to collect characteristics of the respondents related to the context of the study, especially regarding education and employment (such as private practice or public sector, and speciality), and the main location of their practice. The list combines the original EBP$^2$Q demographic data content and the issues specific to the context of the study (Table 1). A template EBP$^2$Q survey form, including demographics, is provided as a S2 File. Responses will be treated anonymously and data will be stored respecting the requirements of personal data integrity and security.

*Sample size*. To achieve the standard error level not higher than 3%, we design the sample size as around 1000. The actual sample will then be standardised according to sex, age, educational level, and living area (Polish voivodship) structure of the whole physiotherapist population, based on the National Registry. The procedure consists of applying the standard formula for the minimum sample size (n) with an assumed level of estimation error (3%) and a confidence level (95%):

$$\frac{70052(1.96^2 \cdot 0,25)}{70052 \cdot 0.03^2 + 1.96^2 \cdot 0.25} \approx 1000,$$

where 1.96 is the value of the normal distribution for the cumulative distribution of $1 - \frac{1-0.95}{2}$ and 0.25 is the constant in the case of an unknown level of a fraction in the population.

Meeting these conditions will allow us to maintain the representativeness of the sample for the Polish physiotherapist population.

**Data collection.** *The questionnaire*. The EBP$^2$Q is a self-reported questionnaire consisting of fifty-eight statements, grouped into five domains–relevance, terminology, confidence, practice, and sympathy, as well as non-domain items of sixteen additional statements addressing other aspects of EBP. The questions are close-ended, with replies ranging in a 5-point Likert scale, some of them reverse-coded. As six separate results are obtained for each respondent, analyses of both separate domains and the profiles of variables are possible. There are no standards for individual domains [20]. The structure and content of the domains of the questionnaire are presented in Table 2.

*Conducting of the survey*. The online survey form, supplemented with an invitation letter, will be distributed via e-mails to all physiotherapists listed in the Registry. Exceptions are people who decline to receive information e-mails from KIF (about 2% of all KIF members at the submission of this paper).

The survey link will be available for three weeks. In the invitation letter, we will present the idea of the project and the purpose of the study, the types of questions that participants could

**Table 1. Items on demographics and professional characteristics of the participants included in the survey.**

| | |
|---|---|
| **personal characteristics** | **gender** |
| | **age** |
| **location of residence** | country of residence |
| | voivodship[1] |
| **location of practice** | rural |
| | town |
| | city |
| **education and formal competencie** | level of education: |
| | technician (vocational school) |
| | bachelor of physiotherapy |
| | master of physiotherapy |
| | specialist of physiotherapy[2] |
| **academic degree or title**[3] | PhD |
| | habilitation[4] |
| | professor |
| **professional experience** | years in profession |
| **work setting** | hospital, clinic |
| | ambulatory |
| | patient's home |
| | residential medical care, nursing home |
| | private practice |
| | health spa |
| | academy |
| | sport club, gym |
| | other |

[1]if Poland country of residence

[2]master of physiotherapy with specialisation

[3]if applicable

[4]the highest scientific degree in the Polish academic system

expect, as well as technical information regarding the time needed to complete the survey (S2 File). Recipients who will not open the first e-mail or will not open the survey form will be emailed with a reminder two weeks after the date of the first dispatch [26]. To reach and encourage everyone eligible, we will additionally distribute the survey messages using the KIF website, newsletter, and social media communication channels.

We will administer the study using the Webankieta web survey platform. It is a Polish design and language platform, dedicated to web-based surveys, and data collection and storage. The survey will be filled out anonymously, and responses cannot be traced back to respondents.

**Data management and analysis.** We will calculate descriptive statistics for the five EBP²Q domains for responses to individual questions and for each domain score, and supplement it with intra-subscale correlation coefficients. We will treat the Likert scores for the questionnaire as ordinal (quazi-quantitative) data. For sociodemographic information, we will calculate descriptive statistics as well.

The association of selected characteristics on the EBP profile of the respondents will be verified with the chi-square test of independence, one-way ANOVA, or Kruskal-Wallis ANOVA depending on the distribution of the EBP²Q scores (providing the Shapiro-Wilk test reveals

**Table 2. The structure and content of the Evidence-Based Practice Profile Questionnaire, EBP²Q [20].**

| Domain | item numbers | description | scale[1] |
|---|---|---|---|
| **Relevance** | 1–14 (14 items) | attitude towards expanding own EBP competence | 1—not at all true |
| | | | 5—very true |
| **Sympathy** | 15–21 (7 items) | attitude towards selected aspects of EBP in work | 1—strongly disagree |
| | | | 5—strongly agree |
| **Terminology** | 22–38 (17 items) | the level of knowledge about the terminology related to scientific research | 1—never heard the term |
| | | | 5—understand and could explain to others |
| **Practice** | 39–47 (9 items) | frequency of use of individual elements of EBP in daily clinical work | 1—never |
| | | | 5—daily |
| **Confidence** | 48–58 (11 items) | confidence in skills related to EBP | 1—not at all confident |
| | | | 5—very confident |
| **non-domain items** | 59–74 (16 items) | other aspects of EBP | 1—strongly disagree |
| | | | 5—strongly agree |

[1] 5-point Likert scale for each item

lack of normal distribution of variables) and/or correlation coefficient significance test. We will apply Pearson or Spearman coefficients for quantitative and ordinal data, respectively, depending on the variables. The threshold for significance will be set at 0.05. The logistic regression will be applied for the identification of the factors which differentiate the importance of three main barriers to EBP implementation, indicated in the questionnaire. All data analyses will be performed using STATISTICA software, v.13.3, StatSoft, Poland.

## Qualitative study (Study 2)

### Background

We find it important to complement the project with a qualitative research study. Qualitative methods offer in-depth, broad and life-immersed perspectives of a phenomenon, which allows for a more comprehensive approach to a complex subject [30]. Therefore, we will expand the quantitative study with a qualitative approach in order to describe and explore the physiotherapists' views and experiences on EBP in the most wide-ranging manner.

### Objectives

We aim at identifying and exploring current views, experiences, beliefs, attitudes, and opinions on EBP amongst physiotherapists in Poland. The focus will be on how physiotherapists actually understand and experience EBP in their work and life, potentially uncovering new insights and viewpoints on the matter.

### Methods

**Design.** We will apply the focus group methodology. We will use the inductive approach, with topics rather than questions. We plan to collect qualitative data having completed Study 1. We will use three focus groups with six to ten physiotherapists each. After the last focus group discussion, the authors will decide whether theoretical saturation is achieved. If not, additional focus groups will then be organised, until no new information or patterns emerge [31, 32]. We expect the interviews to last between 90 and 110 minutes.

After the publication of this protocol, the pilot focus group interview will be conducted first [33]. Next, we will make any necessary amendments to the interview guide, based upon the

pilot interview. The focus group interviews will then be scheduled, taking into consideration specific contextual factors such as the place of the interview, the availability of the research staff and participants, allowing also for enough time after each interview for supervised data transcription and research team feedback sessions.

**Setting and participants.** We will apply purposive sampling, which allows choosing individuals with specific knowledge or experience in a subject of interest [34]. Therefore, we will invite participants based on the network of physiotherapists, connected to the Department of Rehabilitation at the Faculty in Biała Podlaska of the Józef Piłsudski University of Physical Education in Warsaw. It is a rich network of physiotherapists, representing various stages of their professional careers, from graduates to very experienced individuals, working in various settings, and representing ranges of other characteristics (such as age, gender, speciality, sector, work setting). Thus, we aim to ensure intergroup heterogeneity, which will represent a broad spectrum of experiences and contexts [33]. We also want to maintain intragroup homogeneity, in order to encourage open discussions and create a non-threatening group environment. Hence, careful consideration will be given to group dynamics issues such as professional connections, roles and relationships within the focus groups. We will establish appropriate groups based on the characteristics of the participants collected prior to conducting the interviews.

As the quantitative and qualitative studies will be conducted independently, and the data will be managed and analysed separately, we will not consider it as an eligibility criterion if participants willing to take part in the qualitative study will participate in the survey study or not.

**Data collection.** During the focus group interviews, two authors (WK and ZW) will take lead roles as dual moderators [31]. Their roles will include introducing the objectives and the topics of the study, stimulating discussion in the focus groups, and ensuring the appropriate atmosphere. Additionally, reflexive field notes will also be made during the interviews, including the potential reactions, feelings, and non-verbal elements, of both participants and researchers, to present the context of the interviews and provide a more in-depth understanding. This will be done by another research team member (WG), who will serve as an assistant. All interviews will be audiotaped and transcribed verbatim, adding the field noted conducted by the focus group assistant. WG has considerable experience in qualitative research. Additionally, we will invite another experienced qualitative researcher to ensure the rigour of the study's conduct.

The topics will be developed drawing on existing literature on the subject, as well as the personal and professional expertise and experience of the authors. They will be introduced indirectly, in an open-ended manner and focused on the subjects related to experiencing EBP in the participants' practice. This is meant to enable them to naturally engage in posing their own questions and identify priorities within the study's aims and objectives [32]. The key topics will be focused around the following areas:

- understanding of EBP,

- opinions on EBP,

- using EBP in the workplace,

- experiencing EBP in daily practice.

We provide a template table, where we will record and then analyse the themes with illustrative quotes, obtained from the participants (S3 File).

Each focus group interview will begin with opening questions regarding participants' work setting and practice. Participants will be provided with verbal and written information about

the aims of the study and the data collection methods. They will also be notified that they can withdraw at any time. We will hold feedback sessions shortly after each interview.

**Data analysis.** We will analyse the data using the thematic analysis (TA) approach. TA is designed to identify patterns of meaning using the qualitative data, especially useful when exploring data created during focus groups discussions around a specific topic. The analysis will thus consist of data familiarisation, coding, theme development, revision, naming, and report writing [35]. We will apply a staged process of thematic analysis, through an iterative process of meetings and discussions [36]. Three authors (WG, WK, ZW) will first independently analyse the transcribed responses and read them multiple times to familiarise themselves with the content and categorise it in a meaningful way. Then, during the meetings, they will compare the codes and themes, until the shared understanding and consensus is achieved. The themes will be presented narratively, with illustrative data quotes.

## Ethics

The study has received acceptance from the Ministry of Education and Science, Poland, Review Board (SONP/SP/461408/2020), and was further revised by the Ethics Committee of the Józef Piłsudski University of Physical Education in Warsaw. The study adheres to the Declaration of Helsinki.

## Discussion

### Idea and rationale of the study

This is a protocol within a nationwide project aimed at exploring EBP profiles, as well as at identifying barriers and facilitators to the dissemination of EBP amongst physiotherapists and in the entry-level physiotherapy education in Poland [37]. In our view, there is no need to discuss in this paper the paradigm and evolution of EBP and the importance of EBP in general. The established role of EBP in various fields within and outside health care [2, 4, 8, 11], including physiotherapy practice and education [3, 6, 10, 14] are clear. The general need for this study that it is necessary to address this critical issue in terms of practice and research needs and gaps. Several barriers to the use of EBP by physicians and other healthcare professionals [12, 13], and specifically by physiotherapists [14, 15, 21], have been identified, such as an inability to understand statistical data, inadequate support from employers and colleagues, organisational and time constraints, and lack of interest. Understanding potential barriers to EBP is suggested to be the initial step to developing strategies toward successful implementation and dissemination [9, 38].

Our purpose is to explore the issues of knowledge, behaviours, experiences, and attitudes of physiotherapists in Poland towards EBP. A descriptive research study is warranted before analytical investigations and dissemination activities are conducted. More generally, we also aim at promoting and facilitating the EBP culture and use of EBP amongst physiotherapists in Poland. We believe that the most important aspect is the ethical argument within the EBP paradigm. It is unethical to provide patients with suboptimal care, to conduct unneeded research and research leading to unreliable and invalid findings and conclusions. As is wasting resources on cost-inefficient and harmful interventions as well on wasteful science [2, 5, 7, 8]. These issues correspond with the goals of our study and underpin our motivations.

### Study methods and conduct

There is a great variety of tools used to assess EBP knowledge, behaviours and attitudes amongst healthcare professionals [39, 40]. Nonetheless, only the EBP$^2$Q [20] was recently

validated in Polish in a group of nurses and midwives [28, 41]. The EBP$^2$ questionnaire presents good psychometric properties and confirmed reliability and, can be applied to self-assessment of various aspects of EBP competencies by students, lecturers and practitioners [18, 20, 35]. We find it advantageous that a validated Polish translation of this tool is available. The EBP$^2$Q is internationally validated [42, 43] and was used to study EBP profiles of nurses in Poland [28, 41]. This will enable further comparisons.

To conduct the survey, we chose the Webankieta online survey tool as it meets all technical and legal (such as data security) requirements for the planned study, as well as it is equivalent to the SurveyMonkey platform, applied in similar studies [22, 26], while it is provided in the Polish language. It has been used by KIF for other surveys so that both the investigators and recipients are familiar with its use.

As for the qualitative part of the investigation, we decided to choose the focus group methodology. Focus groups encourage interaction and discussion between the participants, in order to enable them to express their personal, subjective, multiple and, at times, even contradictory views, generating potential new ideas and perspectives on the topic studied [30–36]. Qualitative, as well as mixed-method studies, are increasingly utilised to complement quantitative studies in investigating EBP profiles in the context of people's voices, for richer and more in-depth exploration of EBP [33, 36, 44].

Altogether, we believe to collect a meaningful picture of how physiotherapists in Poland see, understand and use EBP.

## Limitations

**Potential biases.**   There may be bias due to the nature of the self-administered questionnaire survey. We are aware of potential non-response bias. There may be a tendency that physiotherapists are more likely to enrol if they are already familiar with the topic. This could overestimate the prevalence estimate found in our sample in the event of a significant non-response. Therefore, we will take preventive steps at the design, implementation and analysis stages of the study, such as repeating invitations to obtain late respondents, providing the accessible format of the survey, monitoring the duration of the survey throughout, as well as adjustment techniques and comparing respondents to non-respondents, if deemed necessary [45]. To achieve the standard error level not higher than 3%, we design the sample size as around 1000. Also, we take into consideration potential phenomena of conscious or unconscious misinterpretations of study findings, or spin bias, especially as we are engaged in the process of EBP and, in that context, we have our attitudes as regards the studied problem [46]. Publishing this registered report protocol is one way of minimizing potential misreporting of methods and results, and misinterpretation. We are also aware of the potential Hawthorne effect, meaning that people could change their behaviour or answer differently when being observed. In contrast, as the study addresses the whole population of interest, and the measures will be recorded identically for all participants, we do not expect selection and ascertainment biases to occur.

**Two separate studies.**   We plan to conduct two individual studies rather than one full mixed-methods study with integrative analyses of the quantitative and qualitative data. Nonetheless, collecting quantitative data, with the use of a recognised, comprehensive questionnaire, as well as a complementary, focus group study aimed at a deeper understanding of the problem, are in our views warranted, needed, relevant, valid, and sufficient for this pioneering study in Poland.

## Dissemination

We plan dissemination activities, such as the website, social media and newsletters messages, and a webinar, to inform physiotherapists as well as to make aware and inspire policymakers

and other stakeholders to advance the dissemination of EBP amongst physiotherapists. Through that process, we also aim to improve physiotherapy curricula in Poland.

## Supporting information

**S1 File.**
(PDF)

**S2 File.**
(PDF)

**S3 File.**
(PDF)

## Acknowledgments

We would like to thank Mr Robert Grantham for peer-reviewing the text for English language soundness, and Mrs Anna Kudelska-Huk for her involvement in the administration and supervision of the project.

## Author Contributions

**Conceptualization:** Maciej Płaszewski.

**Data curation:** Maciej Płaszewski, Weronika Krzepkowska, Weronika Grantham, Zbigniew Wroński, Hubert Makaruk.

**Formal analysis:** Maciej Płaszewski, Weronika Grantham, Joanna Trębska.

**Funding acquisition:** Maciej Płaszewski, Zbigniew Wroński, Hubert Makaruk.

**Investigation:** Weronika Krzepkowska, Weronika Grantham, Zbigniew Wroński, Hubert Makaruk.

**Methodology:** Maciej Płaszewski, Weronika Krzepkowska, Weronika Grantham.

**Project administration:** Maciej Płaszewski, Weronika Krzepkowska, Zbigniew Wroński, Hubert Makaruk.

**Resources:** Maciej Płaszewski, Zbigniew Wroński, Hubert Makaruk.

**Supervision:** Maciej Płaszewski.

**Validation:** Maciej Płaszewski, Weronika Krzepkowska, Weronika Grantham, Zbigniew Wroński, Hubert Makaruk, Joanna Trębska.

**Visualization:** Maciej Płaszewski, Weronika Krzepkowska.

**Writing – original draft:** Maciej Płaszewski, Weronika Krzepkowska, Weronika Grantham.

**Writing – review & editing:** Maciej Płaszewski, Weronika Krzepkowska, Weronika Grantham, Zbigniew Wroński, Hubert Makaruk, Joanna Trębska.

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
