## [Decision Letter · Decision Letter 0]

28 Sep 2021

PONE-D-21-25941Knowledge, behaviours and attitudes towards Evidence-Based Practice amongst physiotherapists in Poland.  A nationwide cross-sectional survey and focus group study protocolPLOS ONE

Dear Dr. Płaszewski,

Thank you for submitting your manuscript to PLOS ONE. After careful consideration, we feel that it has merit but does not fully meet PLOS ONE’s publication criteria as it currently stands. Therefore, we invite you to submit a revised version of the manuscript that addresses the points raised during the review process. Please submit your revised manuscript by Nov 12 2021 11:59PM. If you will need more time than this to complete your revisions, please reply to this message or contact the journal office at plosone@plos.org. Please include the following items when submitting your revised manuscript:A rebuttal letter that responds to each point raised by the academic editor and reviewer(s). You should upload this letter as a separate file labeled 'Response to Reviewers'.A marked-up copy of your manuscript that highlights changes made to the original version. You should upload this as a separate file labeled 'Revised Manuscript with Track Changes'.An unmarked version of your revised paper without tracked changes. You should upload this as a separate file labeled 'Manuscript'.

We look forward to receiving your revised manuscript.

Kind regards,

Ramune Jacobsen

Academic Editor

PLOS ONE

Journal Requirements:

Reviewers' comments:

Reviewer's Responses to Questions

**Comments to the Author**

1. Does the manuscript provide a valid rationale for the proposed study, with clearly identified and justified research questions?

Reviewer #1: Partly

Reviewer #2: Partly

Reviewer #3: Yes

Reviewer #4: Yes

2. Is the protocol technically sound and planned in a manner that will lead to a meaningful outcome and allow testing the stated hypotheses?

Reviewer #1: Partly

Reviewer #2: Partly

Reviewer #3: Yes

Reviewer #4: Yes

3. Is the methodology feasible and described in sufficient detail to allow the work to be replicable?

Reviewer #1: Yes

Reviewer #2: Yes

Reviewer #3: No

Reviewer #4: Yes

4. Have the authors described where all data underlying the findings will be made available when the study is complete?

Reviewer #1: No

Reviewer #2: Yes

Reviewer #3: Yes

Reviewer #4: Yes

5. Is the manuscript presented in an intelligible fashion and written in standard English?

Reviewer #1: Yes

Reviewer #2: Yes

Reviewer #3: Yes

Reviewer #4: Yes

6. Review Comments to the Author

You may also provide optional suggestions and comments to authors that they might find helpful in planning their study.

Reviewer #1: This study protocol describes two planned studies (one quantitative and one qualitative) investigating evidence-based practice among physiotherapists in Poland. In general, I think this is a clear, concise, and well thought out protocol. I have a few questions and suggestions that may improve the manuscript.

1. The introduction could benefit from a little more detail on the concept of evidence-based practice and examples of how this is integrated into physiotherapy.

2. The motivation for conducting these studies could be a bit better motivated. That this exact study has not been conducted in Poland is not enough.

3. The study seems very descriptive, which is ok if this is the level of information that is necessary, but perhaps the aims could be phrased as specific research questions so that it is clearer to the reader exactly what the questions these studies will try to answer.

4. It would also be useful for the authors explain how the research questions will be used. What are the goals and who will be the end users of the information gleaned from these studies?

5. The protocol could benefit from another language/grammatical reading. One example is the last sentence of the first paragraph: “It is unethical to deliver suboptimal, ineffective or dangerous, or cost-ineffective interventions to patients and clients, as unethical is waste in unneeded and flawed research.” To me, this sentence is unclear and difficult to follow.

Reviewer #2: Manuscript Number: PONE-D-21-25941

Title: Knowledge, behaviours and attitudes towards Evidence-Based Practice amongst physiotherapists in Poland. A nationwide cross-sectional survey and focus group study protocol

Comment 1: this is research proposal why don’t you consider its publication after study was conducted?

Comment 2: on page 2 at the end of first paragraph of your abstract you have mentioned as this title/issue was addressed in other parts of the world. So why don’t you use already existed evidences or why not consider doing systematic review and meta- analysis from these evidences.

Comment 3: on page 2 under methods of abstract you have planned to use online survey. One big problem of this method was respondent may update him/her self before responding and may discuss with others. So how you will overcome these problems? You also mentioned use of purposive sampling. Did you think purposive sampling can help in assuring representativeness? Why don’t you consider sampling methods like stratified sampling techniques?

Comment 4: on page 3 just above objectives you have said there is no same study in Poland. Why did you take this as justification to repeat the study? Is there any difference you have found between Poland and other countries physiotherapists?. Add more strong justification to repeat the study.

Comment5: on page 5 under sample size you just plan to take 1000 participants. but you better show clear procedure on how you calculate your sample size. So it help future researcher in replication of the study.

Comment 6: on page 6 under Data management and analysis you have planned to run descriptive statistics. But you should specify what types of descriptive statistics for groups of variables (Example for continuous data and for categorical data).

Comment 7: on page 8 under discussion you wrote a lot of evidence regarding evidence based practice. Did you think you will find different results despite you didn’t mention weakness of previous studies.

Comment 8: on page 9 under limitation you put anticipated limitation. But in reality what you think may not come problematic. Please consider writing limitation after completion of the study.

Comment: 9 on page 10 under author contribution you write some contribution which will expect if study was already conducted (e.g Formal analysis: Maciej Płaszewski, Weronika Grantham, Joanna Trębska). Please revise.

Reviewer: Ahmednur Adem Aliyi

Email: ahmedhariro@gmail.com

Reviewer #3: This research protocol is well design outlying the planning of the authors’ study. It will require minor revision as attached.

Reviewer #4: This is an interesting study and the authors have collected a unique dataset using cutting edge methodology. The paper is generally well written and structured. Overall, this is a clear, concise, and well-written manuscript. The introduction is relevant. Sufficient information about the previous study findings is presented for readers to follow the present study rationale and procedures. The methods are generally appropriate. Overall, the results are clear

7. PLOS authors have the option to publish the peer review history of their article (what does this mean?). If published, this will include your full peer review and any attached files.

Reviewer #1: No

Reviewer #2: No

Reviewer #3: No

Reviewer #4: **Yes: **MAHMOUD AL-MASAEED

---

## [Author Response · Author response to Decision Letter 0]

9 Nov 2021

Dear Reviewers, 

We are grateful for taking the time to support us with expert feedback and valuable input to our manuscript. We are thankful for giving us this opportunity to improve our submission. As part of the submission process, we have provided detailed responses both to the Academic Editor’s and each of the Reviewers’ comments.

Response to the Academic Editor

We greatly appreciate your assistance and contribution to our report. We amended the manuscript in accordance with your suggestions.

Academic Editor’s comment:

 Please ensure that your manuscript meets PLOS ONE's style requirements, including those for file naming. The PLOS ONE style templates can be found at https://journals.plos.org/plosone/s/file?id=wjVg/PLOSOne_formatting_sample_main_body.pdf and https://journals.plos.org/plosone/s/file?id=ba62/PLOSOne_formatting_sample_title_authors_affiliations.pdf

Our response:

Thank you for this suggestion. We have prepared the manuscript accordingly.

Academic Editor’s comment:

 We note that the grant information you provided in the ‘Funding Information’ and ‘Financial Disclosure’ sections do not match. When you resubmit, please ensure that you provide the correct grant numbers for the awards you received for your study in the ‘Funding Information’ section.

Our response:

We carefully checked the grant award numbers in the ‘Financial disclosure’ and ‘Funding Information’ sections and we confirmed that the grant numbers match; however, as there are some wording differences (‘award’ and ‘grant’), we now provide the same text in the manuscript and the online submission form.

Academic Editor’s comment:

Our response:

We have included the ethics statement in the Methods section (as a separate subsection, at the end of the section, so that the statement corresponds to both the quantitative and qualitative parts of the study). We have changed the ‘Ethics and dissemination’ section to the ‘Dissemination’ section. We have moved the subsection to the end part of the ‘Discussion’ section so that it is a part of the main manuscript. And we have added a more in-depth explanation regarding planned dissemination activities, following the suggestions of Reviewer 1.

Once again, we would like to thank you for your support.

Response to the Reviewer #1:

Reviewer’s comment:

 The introduction could benefit from a little more detail on the concept of evidence-based practice and examples of how this is integrated into physiotherapy.

Our response:

We thank the Reviewer for this suggestion. We have followed this suggestion, along with the Reviewer’s 3 corresponding advice. We have improved the ‘Introduction’ section accordingly. Please see the revised manuscript, especially the version with tracked changes.

Reviewer’s comment:

 The motivation for conducting these studies could be a bit better motivated. That this exact study has not been conducted in Poland is not enough.

Our response:

We thank the Reviewer for this suggestion. We have added a more detailed motivation and justification for the study at the end of the ‘Introduction’ section. It also follows a corresponding remark from Reviewer 3. Please see the revised manuscript, specifically the version with the track changes.

Reviewer’s comment:

 The study seems very descriptive, which is ok if this is the level of information that is necessary, but perhaps the aims could be phrased as specific research questions so that it is clearer to the reader exactly what the questions these studies will try to answer.

Our response:

Following this important remark, we have listed specific objectives of our study and addressed them in the revised manuscript, especially in the ‘Discussion’ and ‘Introduction’ sections. The manuscript is supplemented with a study questionnaire file (in Polish), and the original, English version of the questionnaire is also available. We believe that these enhancements, along with the qualitative study key topics, listed in the qualitative study part ‘Methods’ section, give more in-depth insights as regards the specific study questions. Please see the revised manuscript, specifically the version with the track changes.

Reviewer’s comment:

 It would also be useful for the authors explain how the research questions will be used. What are the goals and who will be the end users of the information gleaned from these studies?

Our response:

We elaborate on these issues in the ‘Discussion’ section, specifically under the ‘Idea and rationale of the study’ section. Also, there is a separate ‘Dissemination’ section at the end of the manuscript. However, to make our arguments more clear and accessible to the reader, we rearranged and improved both sections, according to this significant comment. We moved the ‘Dissemination’ part to the end of the ‘Discussion’ section so that the manuscript is now, we hope, more concise. We also address the goals of the study in the improved ‘Introduction’ section, following the Reviewer’s suggestions. Please see the revised manuscript, specifically the version with the track changes.

Reviewer’s comment:

 The protocol could benefit from another language/grammatical reading. One example is the last sentence of the first paragraph: “It is unethical to deliver suboptimal, ineffective or dangerous, or cost-ineffective interventions to patients and clients, as unethical is waste in unneeded and flawed research.” To me, this sentence is unclear and difficult to follow.

Our response:

We have rewritten the indicated sentence. We also checked the whole manuscript for English language quality, with help from a native English speaker, knowledgeable of scientific writing.

Reviewer’s answer to the review question 4:

4. Have the authors described where all data underlying the findings will be made available when the study is complete?

Reviewer’s answer: no

Our response:

We supplemented the original data availability statement provided in the manuscript with the information regarding the anticipated availability of data from the completed study. It now matches, in the ‘Data availability statement’ section of the online submission form, as it has been included in the original submission.

We would like to thank the Reviewer for analysing our paper and for very helpful and constructive comments and suggestions. We hope that our responses and the revised manuscript are complete and sound.

Response to the Reviewer #2:

Reviewer’s comment:

Comment 1: this is research proposal why don’t you consider its publication after study was conducted?

Our response:

We took the advantage of the PLoS policy of open science and the opportunity for publishing full reports of study protocols (the Registered Report Protocol type of study, https://journals.plos.org/plosone/s/what-we-publish#loc-registered-reports) so that the investigation is transparent, and we follow the rule of presenting the protocol openly to ensure scientific rigour and fairness of the final report. Publishing study protocols is encouraged for a number of reasons (please see the PLoS ONE, https://journals.plos.org/plosone/s/submission-guidelines#loc-registered-reports, or BMJ, https://authors.bmj.com/before-you-submit/how-to-write-a-study-protocol/). 

To us, the added value of it is the opportunity to have the study rationale and methods peer-reviewed so that both the study conduct and reporting will gain on quality. 

Reviewer’s comment:

Comment 2: on page 2 at the end of first paragraph of your abstract you have mentioned as this title/issue was addressed in other parts of the world. So why don’t you use already existed evidences or why not consider doing systematic review and meta- analysis from these evidences.

Our response:

One of the main goals of our project is to translate knowledge and to disseminate Evidence-based physiotherapy practice in Poland, among a specific group of people, within specific circumstances. Both undergraduate physiotherapy education, and organisational and policy issues of practice, among other contextual factors, are in Poland very different from many other countries. The data from, and findings of this study, are needed for that process. We first need to conduct descriptive studies in this particular population. The contextual factors are crucial so that we cannot rely on indirect evidence from other contexts. 

The idea of a research synthesis study is interesting. In our view, it needs a thorough consideration of which research synthesis study method would be accurate for such heterogeneous data, with studies conducted in very much varying Evidence-based cultures, settings, and with different study methods and designs. Also, cross-sectional and qualitative study designs are not typically eligible for systematic reviews, especially as regards meta-analyses. Perhaps a scoping review study could be a good starting point for mapping the evidence of that issue. It could help to project the findings of our study in a more systematic perspective.

Reviewer’s comment:

Comment 3: on page 2 under methods of abstract you have planned to use online survey. One big problem of this method was respondent may update him/her self before responding and may discuss with others. So how you will overcome these problems? You also mentioned use of purposive sampling. Did you think purposive sampling can help in assuring representativeness? Why don’t you consider sampling methods like stratified sampling techniques?

Our response:

We agree that a survey method may be vulnerable to recall bias, or Hawthorne effect, among other potential biases, connected to this type of study. Therefore, we address these issues in the ‘Limitations’ part of the study, which is improved according to the Reviewers’ suggestions and advice. Nonetheless, both the survey designs and the online methods of questionnaire distribution and data collection are standard. We are taking advantage of the EBP2 questionnaire, with its good scientific properties, with the availability of Polish validated translation. We are not able to control the fidelity of the responses. Nonetheless, it could not have been checked even if the survey would have been conducted in-person, with the investigator. There are no open questions such as an explanation of a term or definition in the questionnaire.

As regards purposive sampling, we chose this method for the focus group, qualitative study part of our endeavour, as standard in this type of study (Creswell JW, Plano Clark VL. Designing and conducting mixed methods research. Thousand Oaks, CA: SAGE Publications; 2017). We elaborate on this issue in the ‘Methods’ section of Study 2 (‘A focus group approach will be applied for Study 2, with purposive sampling to achieve a representative picture of physiotherapists with respect to the setting, specialty, seniority, educational degrees, and age’), as well as we indicate it in the ‘Abstract’. 

Reviewer’s comment:

Comment 4: on page 3 just above objectives you have said there is no same study in Poland. Why did you take this as justification to repeat the study? Is there any difference you have found between Poland and other countries physiotherapists?. Add more strong justification to repeat the study.

Our response:

Please see our explanation to your first comment. The Evidence-based culture in Poland, on many – individual, organisational, educational, policy levels, is specific and different, in many ways, from other countries and cultures. There is no one, uniform standard of physiotherapy education and conduct. Our study is not a repetition of any other studies (this is one of the reasons why we would like to publish the protocol of our study). The issues of knowledge translation and dissemination of Evidence-based practice are crucial and most important in terms of overcoming the knowledge-to-practice gaps, and it must be tailored to local contexts. Please see e.g. the World EBHC Day 2021 initiative (https://worldebhcday.org/) on how infodemic is being fought throughout the world, in different countries, cultures and settings, to different people, and how Evidence-based health care is important in this subject matter, with the very importance of local, regional and national initiatives. We cannot just rely on earlier studies and data collected in various settings.

The core justification for our study is the need for data collection and for researching the Evidence-based culture amongst physiotherapists in Poland to facilitate further developments such as EBP dissemination and knowledge translation activities. We indicate it in the paper – both in the introduction and in the discussion. In the ‘Objectives’ subsection, we provide the general and specific aims of the study and the whole project. We indicate that ‘The overall aim of the project is to improve and facilitate the process of the dissemination of EBP in Polish physiotherapy’. In the ‘Discussion’ section, we formulate a general justification for the study, where the lack of such study data for Poland is one issue, while the other, regarding the need for study findings, is the issue of critical practice and research gaps: ‘The general need for this study is that no such study was so far conducted in Poland and that it is necessary to address this critical issue in terms of practice and research needs and gaps’.

Reviewer’s comment:

Comment 5: on page 5 under sample size you just plan to take 1000 participants. but you better show clear procedure on how you calculate your sample size. So it help future researcher in replication of the study.

Our response:

The procedure consists of applying the standard formula for the minimum sample size (n) with an assumed level of estimation error (here 3%) and a confidence level (here 95%):

(70052 (〖1.96〗^2∙0,25))/(70052∙〖0.03〗^2+〖1.96〗^2∙0.25)≈1000

1.96 is the value of the normal distribution for the cumulative distribution of 1-(1-0.95)/2 ; 0.25 is the constant in the case of an unknown level of a fraction in the population. 

We included this explanation in the revised paper. Please also see our response to a corresponding question of Reviewer 3. We hope that we have sufficiently addressed these questions. Please see the revised manuscript, specifically the version with the track changes.

Reviewer’s comment:

Comment 6: on page 6 under Data management and analysis you have planned to run descriptive statistics. But you should specify what types of descriptive statistics for groups of variables (Example for continuous data and for categorical data).

Our response:

The use of specific descriptive statistics depends, among others, on the type of a variable and the measurement scale in which the observation results are recorded (in this case, responses to the questionnaire). The most commonly used structure indicators can be calculated for all kinds of variables, both qualitative and quantitative (all measurement scales). The same applies to the dominant (but only for the variables that make up the unimodal series). For ordinal variables (all responses with 5-point Likert scale) the best measure of the average level is the median, but it is allowed to use the arithmetic mean for comparative purposes (e.g. verification of the hypothesis that the level of education is a feature differentiating the answer to a specific question). The selection of dependency measures also depends on the type of the variable: the Pearson coefficient is used for quantitative variables, the Spearman coefficient for ordinal variables, and for nominal features measures based on chi-square statistics.

Reviewer’s comment:

Comment 7: on page 8 under discussion you wrote a lot of evidence regarding evidence based practice. Did you think you will find different results despite you didn’t mention weakness of previous studies. 

Our response:

The individual studies from various countries and settings provide differing findings. Our aim is to conduct a study relevant to the Polish context. An evidence synthesis or critical appraisal of other studies is not our aim in this investigation.

Reviewer’s comment:

Comment 8: on page 9 under limitation you put anticipated limitation. But in reality what you think may not come problematic. Please consider writing limitation after completion of the study.

Our response:

We will discuss the limitations in our report of the completed study. Here we discussed limitations to ensure the soundness and clarity of the protocol. We concentrated on the study design limitations and potential (anticipated) study conduct limitations, not on the limitations of the study findings (which we will address in the study report). Especially, we indicated potential biases which we are aware of. As this is a methods report, we concentrated on those issues. This part of the paper is also amended, following the suggestions of the Reviewers. Please see the revised manuscript, specifically the version with the track changes.

Reviewer’s comment:

Comment: 9 on page 10 under author contribution you write some contribution which will expect if study was already conducted (e.g Formal analysis: Maciej Płaszewski, Weronika Grantham, Joanna Trębska). Please revise.

Our response:

We followed the definitions of the study contributor roles as provided by the Journal (https://journals.plos.org/plosone/s/authorship#loc-author-contributions), and considered all current and future roles in the conduct of the study and preparation of the reports. Therefore, we indicated the subsequent roles. The ‘Formal analysis’ role is defined as ‘Application of statistical, mathematical, computational, or other formal techniques to analyze or synthesize study data’. Therefore, we listed those of us who have contributed to the data analysis methods conceptualisation and reporting (which is done in the submitted manuscript under its relevant sections. For the same reasons, we indicated those of us who contribute within the ‘Investigation’ and ‘Data curation’ roles. We acknowledge the Journal’s policy regarding authorship. We consulted other registered report protocols, published at PLoS One, and some of them also indicate authors’ roles for the whole study process (e.g. https://journals.plos.org/plosone/article?id=10.1371/journal.pone.0253950), while others not (e.g. https://journals.plos.org/plosone/article?id=10.1371/journal.pone.0257223). Therefore, we would like to leave the role descriptions as stated. The responses to the reviewers will be published alongside the paper, providing it is accepted, so that the full information and transparency will be ensured (https://journals.plos.org/plosone/s/revising-your-manuscript). 

We are grateful to the Reviewer for taking his time for a thorough analysis of our paper, and for sharing with us his considerations. We hope that our replies are clear and informative.

Response to the Reviewer #3:

Reviewer’s comment:

Abstract: conclusions should be rewritten to explain the proposed or expected outcome rather than repeating the purpose and objectives of the study under conclusions. 

Our response:

Thank you for this comment and the suggestion. We formulated the conclusions to address the protocol report so that we did not address the outcomes. Also, we had some difficulty in formulating conclusions in a study protocol. There are protocol reports of corresponding investigations with no conclusions in the abstract as well as in the full text (e.g. Mariano AS, Souza NM, Cavaco A, et al. Healthcare professionals’ behavior, skills, knowledge and attitudes on evidence-based health practice: a protocol of cross-sectional study. BMJ Open 2018;8:e018400). On the other hand, others address study rationale as conclusions (e.g. Kapitza C, Lüdtke K, Tampin B, Ballenberger N (2020) Application and utility of a clinical framework for spinally referred neck-arm pain: A cross-sectional and longitudinal study protocol. PLoS ONE 15(12): e0244137). 

Considering your comment, the fact that we have no ‘Conclusions’ section in the full text of our paper, and taking the unclear and unstandardised place of ‘Conclusions’ sections of protocol reports, we decided to delete the ‘Conclusions’ section of the abstract. We believe that this corresponds with your remarks. Please see the revised manuscript, specifically the version with the track changes.

Reviewer’s comment:

Introduction: should be rewritten. The authors quoted 40 references under introduction with only four short paragraphs. 

Our response:

We reconsidered the reference list and deleted a few unnecessary citations. We especially reconsidered refs. 24-29, regarding EBP in physiotherapy ‘….across countries and contexts…’ and refs. 33-40, regarding also EBP studies ‘amongst physiotherapists worldwide’, which were overlapping. We left representative examples and the most current reports so that our statements remain supported with sufficient and informative, but not excessive referencing. We added one reference addressing the definition of EBP. Please see the revised manuscript, specifically the version with the track changes.

Reviewer’s comment:

Start with the definition of EBP and the introduction will benefit from detailed explanation. 

Our response:

This, and the former, correspond with the comments of Reviewer 1. We thank the Reviewer for the suggestions We supplemented the introduction accordingly, adding one most relevant and current definition of EBP. Please see the revised manuscript, specifically the version with the track changes.

Reviewer’s comment:

The authors mentioned at the end of introduction “In contrast, no such research was completed in Poland. This is the first study to address EBP profiles of physiotherapists practicing in Poland”. This is a strong statement and cannot be granted as the authors unaware of unpublished work in this field. I suggest deleting this statement. 

Our response:

Thank you for this advice. Indeed, we did not conduct any formal study to support this statement. Nonetheless, as we have a thorough recognition of the subject matter, and the argument is crucial, we will leave a more careful and legitimate communication, adding the ‘to the best of our knowledge’ account. Please see the revised manuscript, specifically the version with the track changes.

Reviewer’s comment:

Objectives: under specific objectives please add “ To formulate recommendations based on the findings of the study” 

Our response:

Thank you for this suggestion. We improved the text accordingly, inserting a corresponding statement, and including an additional explanation regarding the aims of the study. Please see the revised manuscript, specifically the version with the track changes.

Reviewer’s comment:

Methods: 

Quantitative study: The validity of questionnaire should be obtained from reference 42 and their Cronbach’s alpha and reliability scores should be reported after data collection.

Our response:

Thank you for this suggestion. We will report the Cronbach’s alpha and reliability scores of the questionnaire in our final report.

Reviewer’s comment:

Explain in more details’ inclusion and exclusion criteria of the participants. 

Our response:

In the quantitative part of the study (Study 1), all physiotherapists registered in Poland will be eligible. We described this criterion in detail in the ‘Setting and participants’ section, with explanations regarding such potentially disturbing issues as level of education or nationality. 

As regards the qualitative part (Study 2), we applied the same layout of the manuscript, so that the ‘Setting and participants’ section of Study 2 includes details on focus group interview eligibility criteria. 

Under that section, we also address the issue of the eligibility of participants who would be willing to participate in both studies.

Reviewer’s comment:

What is the proposed or estimated response rate among the participants?

Our response:

The study assumed the minimum sample size of 1000, without estimating the response rate. Please see our reply to your question regarding sample size calculation. Based on surveys previously conducted by our institution in the same population, using the same communication methods and procedures, we do not expect a response below 1000. We will conduct specific actions to ensure a response, as we have described in the 'Conduct of the survey' section.

Reviewer’s comment:

Under setting and participants, second paragraph: the statement “ A small number of physiotherapists ……” How many? Need reference to support it. 

Our response:

Thank you for this suggestion. We deleted the phrase ‘A small number of…’ as it is vague and is not relevant in terms of eligibility criteria. Registered physiotherapists, who had completed their vocational training before 2015, are eligible (providing they meet the remaining criteria), regardless of their number. 

Reviewer’s comment:

Table 1: outline the presence of many confounding factors with focus on work experience and highest qualifications which are related to outcome of KAP. 

Our response:

We have included these remarks in the ‘Discussion’ section, under the limitations of the study. Please see the revised manuscript, specifically the version with the track changes.

Reviewer’s comment:

Under sample size: How the data will be extracted? What will be the sampling technique? What is the sampling equation to calculate sample size? How did the authors come up with around 1000?

Our response:

The procedure consists of applying the standard formula for the minimum sample size (n) with an assumed level of estimation error (here 3%) and a confidence level (here 95%):

(70052 (〖1.96〗^2∙0,25))/(70052∙〖0.03〗^2+〖1.96〗^2∙0.25)≈1000

1.96 is the value of the normal distribution for the cumulative distribution of 1-(1-0.95)/2 ; 0.25 is the constant in the case of an unknown level of a fraction in the population. 

Please also see our response to a corresponding question of Reviewer 2. We hope that we have sufficiently addressed these questions. Please see the revised manuscript, specifically the version with the track changes.

Reviewer’s comment:

Under data collection: what are the dependent and independents variables?

Our response:

Independent variables are all demographic and professional characteristics listed in Table 1, and dependent variables are the responses to the EBP2 questionnaire (Table 2).

Reviewer’s comment:

I have a concern with 5-point Likert scale as it has too many available categories which may obscure rather than define the purpose of the respondent. Therefore, I suggest collapsing into 3 categories across response which may improve the outcome of the analysis (Reference: Grimbeek P., Bryer F., Beamish W., D’Netto M. Stimulating the’Action’ as Participants in Participatory Research. Volume 2. Griffith University; Brisbane, Australia: 2005. Use of data collapsing strategies to identify latent variables in CHP questionnaire data; p. 125.).

Furthermore, each response in the scale should be assigned a numerical value so that parametric statistics can be used for Likert data. Higher score indicates positive response. 

Our response:

We will consider your suggestions in the analysis. However, in the EBP2Q, with the procedure which we will need to follow, the questions have the replies ranged in a 5-point Likert scale, some of them reverse-coded. We, therefore, address this issue in the ‘Data collection’ part of the paper, including in Table 2. The variables resulting from the answers provided in the questionnaire will be treated as quazi-measurable / quazi-quantitative (assigning the answers with numerical values, as indicated in Table 2, and in the ‘Data management and analysis’ section). Nonetheless, in the statistical analysis, it may turn out that some of the responses will be aggregated to a 3-point scale. It is better, however, that we have the possibility to reduce the number of variants. 

Reviewer’s comment:

Qualitative study: this will definitely complement the quantitative study; however, it may be a problematic during COVID-19 pandemic. If this is the case, those conducting the interview should implement physical distance restrictions and wear face mask during the interview to overcome this problem. 

Our response:

We will conduct the interview in a responsible way. We acknowledge, implement and promote all safety restrictions and responsible behaviours as regards COVID-19 risks.

Reviewer’s comment:

What is the duration of the interview? When will it take place? Over span of how long? 

Our response:

The duration of the interview is described in the ‘Design’ part of the ‘Methods’ section of Study 2. Following your suggestion, we added the information about when and over span of how long the interviews will take place. Please see the revised manuscript.

Reviewer’s comment:

The one conducting interview should have experience in qualitative research.

Experts in qualitative research should supervised data collection and hold feedback sessions shortly after the interview.

Our response:

WG, one of the co-authors, has considerable experience in qualitative research. Additionally, we will invite another experienced qualitative researcher to ensure the rigour of the study conduct. We added such information in the manuscript.

Reviewer’s comment:

All interviews should be recorded and transcribed in full to verify the accuracy of responses.

Our response:

This procedure is described in the ‘Data collection’ part of the ‘Methods’ section of Study 2.

Reviewer’s comment:

Two members of the research team independently should analyze the transcribed responses and read them on multiple times to familiarize self with the contents and categorize it in a meaningful way. 

All these points should be explained and incorporated under the design of qualitative study. 

Our response:

This issue is described under the ‘Data analysis’ section. Nonetheless, to improve accuracy, we included additional information and explanations addressing these important methods issues. Thank you for your suggestions.

Reviewer’s comment:

Limitations of the study: should explain in details and logical ways. Personal, recall, misinterpretation biases are possibilities in this study.

Our response:

We added two subheadings (‘Potential biases’ and ‘Two separate studies’), to clarify to the readers the two potential areas of study limitations – potential biases connected to these study designs and their conduct, as well as separate conducting of the quantitative and qualitative studies.

We improved the discussion of potential biases with a more specific addressing to potential biases and potential misinterpretation. As we are researching current EBP profiles of our participants, and this is not a retrospective cohort or case-control study, we will refrain from discussing recall bias as it is defined by the CEBM Oxford [Catalogue of Bias Collaboration, Spencer EA, Brassey J, Mahtani K. Recall bias. In: Catalogue of Bias 2017. https://www.catalogueofbiases.org/biases/recall-bias].

Reviewer’s comment:

References: The manuscript presents several references which are outdated (1 and 44), and some are irrelevant. The references should be revised, updated, and only used if it is relevant to the study. It also should follow the guidelines of the journal.

Our response:

We followed the Reviewer’s suggestions. As the reference 1, however, is the first full text, classical paper defining the paradigm of EBM, and is needed for our introduction – ‘It is now thirty years since the Evidence-Based Medicine Working Group coined the term (…)’, we would like to leave this reference as it is. However, we deleted ref. 3, as its content is halfway refs 1 and 2, and factually does not support any additional statements in the paper. Please, see also our reply to the comment regarding the ‘Introduction’ section, which also addresses the reference list.

We are grateful for taking your time to support us with your in-depth, expert feedback which did help us to improve our manuscript.

Response to the Reviewer #4:

Reviewer’s comment:

This is an interesting study and the authors have collected a unique dataset using cutting edge methodology. The paper is generally well written and structured. Overall, this is a clear, concise, and well-written manuscript. The introduction is relevant. Sufficient information about the previous study findings is presented for readers to follow the present study rationale and procedures. The methods are generally appropriate. Overall, the results are clear.

Our response:

We would like to thank the Reviewer for taking their time and effort and analysing our manuscript. We appreciate the positive and encouraging comments.

---

## [Decision Letter · Decision Letter 1]

4 Dec 2021

PONE-D-21-25941R1Knowledge, behaviours and attitudes towards Evidence-Based Practice amongst physiotherapists in Poland.  A nationwide cross-sectional survey and focus group study protocolPLOS ONE

Dear Dr. Płaszewski,

Thank you for submitting your manuscript to PLOS ONE. After careful consideration, we feel that it has merit but does not fully meet PLOS ONE’s publication criteria as it currently stands. Therefore, we invite you to submit a revised version of the manuscript that addresses the points raised during the review process.

We look forward to receiving your revised manuscript.

Kind regards,

Ramune Jacobsen

Academic Editor

PLOS ONE

Journal Requirements:

Reviewers' comments:

Reviewer's Responses to Questions

**Comments to the Author**

1. Does the manuscript provide a valid rationale for the proposed study, with clearly identified and justified research questions?

Reviewer #1: Yes

Reviewer #2: Yes

Reviewer #3: Yes

Reviewer #4: Yes

2. Is the protocol technically sound and planned in a manner that will lead to a meaningful outcome and allow testing the stated hypotheses?

Reviewer #1: Yes

Reviewer #2: Yes

Reviewer #3: Yes

Reviewer #4: Yes

3. Is the methodology feasible and described in sufficient detail to allow the work to be replicable?

Reviewer #1: Yes

Reviewer #2: Yes

Reviewer #3: Yes

Reviewer #4: Yes

4. Have the authors described where all data underlying the findings will be made available when the study is complete?

Reviewer #1: Yes

Reviewer #2: Yes

Reviewer #3: Yes

Reviewer #4: Yes

5. Is the manuscript presented in an intelligible fashion and written in standard English?

Reviewer #1: Yes

Reviewer #2: Yes

Reviewer #3: Yes

Reviewer #4: Yes

6. Review Comments to the Author

You may also provide optional suggestions and comments to authors that they might find helpful in planning their study.

Reviewer #1: The authors have done a good job responding to the reviewers and editing the study protocol accordingly. I only have a few final suggestions.

While the author’s have partially addressed the motivation behind the study beyond that it is the first of its kind in Poland, I still think the manuscript could benefit from a little bit stronger motivation. For example, on page 3 in the statement “to the best of our knowledge, no such research was completed in Poland” could be enhanced with something like “and there are unique factors of this context which may be important” etc. Also, the abstract still lists the study being in Poland as the only real motivation.

There are still some grammatical mistakes and sentences which could be edited.

The authors bring up the Hawthorne effect in the limitations section. It could be useful to the reader to briefly explain what this is (that is, that people change their behavior or answer differently when being observed).

Finally, the abstract now only consists of objectives and methods, I understand this because it is a protocol, but would it be possible to just have an unstructured abstract instead?

Reviewer #2: Dear author of this study,

I will not pass every effort you made in responding to comments and correction you made in your documents.

But I believe you should modify your last objective which say “to identify the facilitators and barriers in the implementation of EBP they experience in their everyday practice” . Because you planned to use crossectional study (survey) which can’t tell us direction of relationships. I recommend you to modify like “ to identify factors associated with EBP among physiotherapist”

Reviewer #3: The required corrections have been made and the manuscript has been improved scientifically. The authors responded to my comments and provided valuable information which are important for the readers. Thank you.

Reviewer #4: Thanks for addressing the comments and i can see the improvement in the manuscript titled : Knowledge, behaviours and attitudes towards Evidence-Based Practice amongst physiotherapists in Poland. A nationwide cross-sectional survey and focus group study protocol

7. PLOS authors have the option to publish the peer review history of their article (what does this mean?). If published, this will include your full peer review and any attached files.

Reviewer #1: **Yes: **Melody Claire Almroth

Reviewer #2: No

Reviewer #3: No

Reviewer #4: **Yes: **Mahmoud Al-Masaeed

---

## [Author Response · Author response to Decision Letter 1]

10 Dec 2021

Dear Reviewers, 

We would like to thank you for the analysis of our revised manuscript and for your valuable feedback. Please find below our responses to your comments.

Response to the Reviewer #1:

Reviewer’s comment:

The authors have done a good job responding to the reviewers and editing the study protocol accordingly. I only have a few final suggestions.

While the author’s have partially addressed the motivation behind the study beyond that it is the first of its kind in Poland, I still think the manuscript could benefit from a little bit stronger motivation. For example, on page 3 in the statement “to the best of our knowledge, no such research was completed in Poland” could be enhanced with something like “and there are unique factors of this context which may be important” etc. Also, the abstract still lists the study being in Poland as the only real motivation.

Our response:

We thank the Reviewer for this suggestion. We added a more detailed justification to the sentence you indicated. We also added an explanatory statement in the abstract, in the ‘Objectives’ section. 

Reviewer’s comment:

There are still some grammatical mistakes and sentences which could be edited.

Our response:

The manuscript was double-checked by our native English colleague. He still found some inaccuracies and they are now rewritten. Thank you for this remark.

Reviewer’s comment:

The authors bring up the Hawthorne effect in the limitations section. It could be useful to the reader to briefly explain what this is (that is, that people change their behavior or answer differently when being observed).

Our response:

Thank you for this suggestion. We have added a short explanation, following the Hawthorne effect mention. 

Reviewer’s comment:

Finally, the abstract now only consists of objectives and methods, I understand this because it is a protocol, but would it be possible to just have an unstructured abstract instead?

Our response:

Thank you for this suggestion, but we would like to leave the structured layout of the protocol, as standard fot PLoS ONE, including registered report protocols, such as Spitzer L, Mueller S (2021) Registered Report Protocol: Survey on attitudes and experiences regarding preregistration in psychological research. PLoS ONE 16(7):e0253950 and Adams D, Malone S, Simpson K, Tucker M, Rapee RM, Rodgers J, et al. (2021) Protocol for a longitudinal study investigating the role of anxiety on academic outcomes in children on the autism spectrum. PLoS ONE 16(9): e0257223, which also have no ‘Conclusions’ sections.

Response to the Reviewer #2:

Reviewer’s comment:

Dear author of this study,

I will not pass every effort you made in responding to comments and correction you made in your documents.

But I believe you should modify your last objective which say “to identify the facilitators and barriers in the implementation of EBP they experience in their everyday practice” . Because you planned to use crossectional study (survey) which can’t tell us direction of relationships. I recommend you to modify like “ to identify factors associated with EBP among physiotherapist”

Our response:

We thank the Reviewer for this remark. We have rewritten the objective so that it is now not suggesting an explanatory, rather than a descriptive study. 

We would like to thank the Reviewer 3 and the Reviewer 4 for analysing our revised manuscript and for their replies. 

We are grateful to the Reviewers and to the Academic Editor for their valuable input to our manuscript. Once again, we do appreciate your expert support. Our paper, and our study, have considerably improved through this peer-review process.

---

## [Decision Letter · Decision Letter 2]

4 Jan 2022

PONE-D-21-25941R2Knowledge, behaviours and attitudes towards Evidence-Based Practice amongst physiotherapists in Poland.  A nationwide cross-sectional survey and focus group study protocolPLOS ONE

Dear Dr. Płaszewski,

Thank you for submitting your manuscript to PLOS ONE. After careful consideration, we feel that it has merit but does not fully meet PLOS ONE’s publication criteria as it currently stands. Therefore, we invite you to submit a revised version of the manuscript that addresses the points raised during the review process.

We look forward to receiving your revised manuscript.

Kind regards,

Ramune Jacobsen

Academic Editor

PLOS ONE

Journal Requirements:

Additional Editor Comments:

Thank for your good work on this manuscript/protocol. The reviewers are happy, and I am willing to accept it after some minor/editing revisions:

1) editing the abstract: deleting duplications of methods in the online version, and adding something on the results (i.e. how they will be presented) and discussion (i.e. limitations);

2) making presentation plan consistent for the survey and the qualitative study (i.e. objective, methods, results); please note, I believe that it’s easier for the reader to see the plan for the analysis and results presentation (e.g. empty tables) separately, as it would be in a research article.

Reviewers' comments:

Reviewer's Responses to Questions

**Comments to the Author**

1. Does the manuscript provide a valid rationale for the proposed study, with clearly identified and justified research questions?

Reviewer #1: Yes

Reviewer #2: Yes

Reviewer #3: Yes

Reviewer #4: Yes

2. Is the protocol technically sound and planned in a manner that will lead to a meaningful outcome and allow testing the stated hypotheses?

Reviewer #1: Yes

Reviewer #2: Yes

Reviewer #3: Yes

Reviewer #4: Yes

3. Is the methodology feasible and described in sufficient detail to allow the work to be replicable?

Reviewer #1: Yes

Reviewer #2: Yes

Reviewer #3: Yes

Reviewer #4: Yes

4. Have the authors described where all data underlying the findings will be made available when the study is complete?

Reviewer #1: Yes

Reviewer #2: Yes

Reviewer #3: Yes

Reviewer #4: Yes

5. Is the manuscript presented in an intelligible fashion and written in standard English?

Reviewer #1: Yes

Reviewer #2: Yes

Reviewer #3: Yes

Reviewer #4: Yes

6. Review Comments to the Author

You may also provide optional suggestions and comments to authors that they might find helpful in planning their study.

Reviewer #1: The authors have addressed my final concerns and the manuscript, in my opinion, is ready for publication.

Reviewer #2: The manuscript was written with great effort. They respond to reviewer comments point by point. Thanks.

Reviewer #3: The manuscript looks now much better. The required corrections have been made and the manuscript has been improved scientifically. The authors responded to my comment and provided valuable information which are important for the readers. I suggest accepting for publication at your respectful journal

Reviewer #4: Dear Atheros

This is an interesting study and the authors have collected a unique dataset using cutting edge methodology. The paper is generally well written and structured. Overall, this is a clear, concise, and well-written manuscript. The introduction is relevant. Sufficient information about the previous study findings is presented for readers to follow the present study rationale and procedures. The methods are generally appropriate. Overall, the results are clear

7. PLOS authors have the option to publish the peer review history of their article (what does this mean?). If published, this will include your full peer review and any attached files.

Reviewer #1: No

Reviewer #2: No

Reviewer #3: No

Reviewer #4: **Yes: **MAHMOUD AL-MASAEED

---

## [Author Response · Author response to Decision Letter 2]

7 Feb 2022

Dear Reviewers,

Thank you for your final analysis of our manuscript. We are thankful for your valuable input to this paper and to this project. Thank you.

Thank you for your points you have raised to our revised manuscript. And thank you for the time and effort you have given to the whole process of scientific editing of our submission.

Please find below our elaboration on how we addressed your suggestions in the amended manuscript.

Editor comment:

Editing the abstract: deleting duplications of methods in the online version, and adding something on the results (i.e. how they will be presented) and discussion (i.e. limitations).

Our response:

Based on your suggestions, we have made the following changes:

Both the online, and paper, versions, are now identical.

The Abstract, especially the issue of whether to include Results and Discussion/Conclusions sections in the Abstract had been previously raised by two of the Reviewers and we have already changed the abstract, especially the sections under discussion. Therefore, while preparing this revision, we took into our considerations those comments and suggestions of the Reviewers. Nonetheless, we think that your suggestion is final and decisive in this matter, especially in terms of the Journal’s requirements, so that we have added the Results and Discussion as you have indicated.

The word count for Abstract is 300 in the revised manuscript. 

Editor comment:

Making presentation plan consistent for the survey and the qualitative study (i.e. objective, methods, results); please note, I believe that it’s easier for the reader to see the plan for the analysis and results presentation (e.g. empty tables) separately, as it would be in a research article.

Our response:

Thank you for this suggestion. We have analysed the paper again in the context of the separate presentation of the two legs of our study (the survey, Study 1, and the focus group study, Study 2) and we have decided as follows:

we believe the layout of the paper, as it is already provided, is clear and understandable in terms of the separate description of the objectives and methods of Study 1 and Study 2: we provide separate sections (i.e. QUANTITATIVE STUDY (Study 1); QUALITATIVE STUDY (Study 2)) and subsections to both sections (such as Objectives, Methods, Study design, Data collection and analysis); but, we are aware that reporting two planned studies in one paper is challenging and we do understand your concern, so that we have decided to add to the manuscript a supplementary file – the flow chart of the study, with a graphic presentation of the planned study conduct, with its quantitative and qualitative part; we have added a corresponding statement in the paper, at the beginning of the Methods section, before the description of Study 1 and Study 2; we believe it will serve as additional guide for the readers;

as regards the results and planned analyses, we have provided an additional table – also as a supplementary file – a template table which will be used during the focus group meetings for data (narratives) collection and then for analyses of the qualitative part (Study 2); we have added (in the Data collection section) a relevant statement in the paper – leading to the supplementary file (Supplementary File 3);

as regards the survey part (Study 1) the manuscript includes a demographics table and a EBP2Q structure and content table, which, when used with the template survey (supplementary file – already included, now Supplementary File 2), provides in our view a sufficient guide for the readers if how the data will be collected and then how the results will be presented. 

Again, thank you very much for your support and guidance. We believe that the provided improvements will meet the requirements of the Journal. For a technical pip from the Journal, please see below our reply.

---

## [Editor Report · Decision Letter 3]

14 Feb 2022

Knowledge, behaviours and attitudes towards Evidence-Based Practice amongst physiotherapists in Poland.  A nationwide cross-sectional survey and focus group study protocol

PONE-D-21-25941R3

Dear Dr. Płaszewski,

We’re pleased to inform you that your manuscript has been judged scientifically suitable for publication and will be formally accepted for publication once it meets all outstanding technical requirements.

Kind regards,

Ramune Jacobsen

Academic Editor

PLOS ONE

---

## [Editor Report · Acceptance letter]

21 Feb 2022

PONE-D-21-25941R3 

Knowledge, behaviours and attitudes towards Evidence-Based Practice amongst physiotherapists in Poland.  A nationwide cross-sectional survey and focus group study protocol 

Dear Dr. Płaszewski:

I'm pleased to inform you that your manuscript has been deemed suitable for publication in PLOS ONE. Congratulations! Your manuscript is now with our production department. 

Kind regards, 

on behalf of

Dr. Ramune Jacobsen 

Academic Editor

PLOS ONE